# Closing the Gap: Bridging Peripheral Sensory Nerve Defects with a Chitosan-Based Conduit a Randomized Prospective Clinical Trial

**DOI:** 10.3390/jpm12060900

**Published:** 2022-05-30

**Authors:** Arne Böcker, Martin Aman, Ulrich Kneser, Leila Harhaus, Frank Siemers, Felix Stang

**Affiliations:** 1Department of Hand-, Plastic and Reconstructive Surgery, Burn Center, Department of Plastic and Hand Surgery, University of Heidelberg, BG Trauma Hospital Ludwigshafen, 67071 Ludwigshafen, Germany; martin.aman@bgu-ludwigshafen.de (M.A.); ulrich.kneser@bgu-ludwigshafen.de (U.K.); leila.harhaus@bgu-ludwigshafen.de (L.H.); 2Department of Plastic- and Hand Surgery, Burn Center, Martin-Luther University Halle-Wittenberg, BG Trauma Center Bermannstrost, 06002 Halle, Germany; frank.siemers@bergmannstrost.de; 3Department of Plastic Surgery, University Hospital Schleswig-Holstein, Campus 23562 Luebeck, Germany; felix.stang@uksh.de

**Keywords:** peripheral nerve injuries, nerve gap, nerve surgery, nerve regeneration, nerve conduit, chitosan

## Abstract

**Introduction:** If tensionless nerve coaptation is not possible, bridging the resulting peripheral nerve defect with an autologous nerve graft is still the current gold standard. The concept of conduits as an alternative with different materials and architectures, such as autologous vein conduits or bioartificial nerve conduits, could not replace the nerve graft until today. Chitosan, as a relatively new biomaterial, has recently demonstrated exceptional biocompatibility and material stability with neural lineage cells. The purpose of this prospective randomized clinical experiment was to determine the efficacy of chitosan-based nerve conduits in regenerating sensory nerves in the hand. **Materials and methods****:** Forty-seven patients with peripheral nerve defects up to 26 mm distal to the carpal tunnel were randomized to receive either a chitosan conduit or an autologous nerve graft with the latter serving as the control group. Fifteen patients from the conduit group and seven patients from the control group were available for a 12-month follow-up examination. The primary outcome parameter was tactile gnosis measured with two-point discrimination. The secondary outcome parameters were Semmens Weinstein Monofilament Testing, self-assessed pain, and patient satisfaction. **Results:** Significant improvement (in static two-point discrimination) was observed six months after trauma (10.7 ± 1.2 mm; *p* < 0.05) for chitosan-based nerve conduits, but no further improvement was observed after 12 months of regeneration (10.9 ± 1.3 mm). After six months and twelve months, the autologous nerve graft demonstrated comparable results to the nerve conduit, with a static two-point discrimination of 11.0 ± 2.0 mm and 7.9 ± 1.1 mm. Semmes Weinstein Filament Testing in the nerve conduit group showed a continuous improvement over the regeneration period by reaching from 3.1 ± 0.3 after three months up to 3.7 ± 0.4 after twelve months. Autologous nerve grafts presented similar results: 3.3 ± 0.4 after three months and 3.7 ± 0.5 after twelve months. Patient satisfaction and self-reported pain levels were similar between the chitosan nerve conduit and nerve graft groups. One patient required revision surgery due to complications associated with the chitosan nerve tube. **Conclusion:** Chitosan-based nerve conduits are safe and suitable for bridging nerve lesions up to 26 mm in the hand. Tactile gnosis improved significantly during the early regeneration period, and functional outcomes were similar to those obtained with an autologous nerve graft. Thus, chitosan appears to be a sufficient substitute for autologous nerve grafts in the treatment of small nerve defects in the hand.

## 1. Introduction

Bridging peripheral nerve defects remains a challenging problem in the field of peripheral nerve surgery. As tensionless coaptation of the nerve is of utmost importance for regeneration, any defects require surgical alternatives when direct coaptation is not feasible. As a result, for decades, the autologous nerve graft (ANG) has been the gold standard in the treatment of peripheral nerve injuries (PNI) [1,2].

ANG comes with the burden of donor-side morbidity and limited availability. Alternatives include a common clinical request. One concept in related research, although still lacking clinical data, is the use of different bioartificial nerve conduits. Conduit materials range from autologous materials, such as veins [3] or veins with an inner muscle layer [4], up to bioartificial nerve conduits based on collagen [5] or polylactin acid [6]. Chitosan is a relatively new material for applications in nerve reconstruction. Based on arthropods’ shells, chitosan is widely available in nature and easy to modify for clinical use [7]. With its biocompatibility and potential for biodegradation in the absence of toxic metabolites, chitosan fulfills Schmidt et al. recommendations for a nerve conduit material with the potential to serve as a suitable alternative to ANG [8]. Numerous advantages have been demonstrated, including support for axonal regeneration [9,10], reduction in scar tissue formation [11], and prevention of neuroma formation [12]. Additionally, the first clinical studies chitosan nerve conduits focused on protecting the nerve coaptation site and demonstrated a significantly faster recovery relative to two-point discrimination in a prospective RCT design [13]. In addition to all these potential benefits for nerve recovery, chitosan-bridging peripheral nerve defects continue to be a challenging problem in peripheral nerve surgery. This study aimed to demonstrate the regeneration potential of chitosan-based nerve conduits (CNC) in digital nerve defects compared to the current gold standard: the ANG.

## 2. Materials and Methods

### 2.1. Study Design and Setting

This is a multi-center, randomized, and controlled trial with a parallel-group design. The study was conducted in the trauma centers of the BG Trauma Center Bergmannstrost in Halle (Germany) and the University Hospital Schleswig-Holstein in Luebeck (Germany). The blinded assessment was conducted to compare the conduits’ potential for bridging peripheral nerve defects in the finger to that of the ANG, with an emphasis on tactile gnosis and functional recovery. All subjects provided informed consent for inclusion before they participated in the study. The study was conducted in accordance with the Declaration of Helsinki, and the protocol was approved by the Ethics Committee (DIMDI number 00030916). The primary outcome parameter was the static and dynamic two-point discrimination representative for tactile gnosis. Furthermore, Semmes Weinstein Filament tests for sensory recovery, pain, and patient satisfaction were assessed. All patients included in the study provided informed consent.

### 2.2. Inclusion and Exclusion Criteria

#### 2.2.1. Preoperative Inclusion Criteria

Preoperative inclusion criteria for patients were as follows: lesion between the distal border of the carpal tunnel and the distal finger joint, age between 18 and 65 years, trauma within a time period of six months (including acute trauma), signed informed consent, and clinically complete loss of nerve-related sensitivity.

#### 2.2.2. Intraoperative Inclusion Criteria

Intraoperative inclusion criteria included the confirmation of peripheral nerve injury (PNI) in the presence of a nerve gap that excludes tension-free nerve coaptation. The intraoperative nerve gap must be less than 30 mm long.

#### 2.2.3. Preoperative Exclusion Criteria

Preoperative exclusion criteria included the following: amputated finger, avascular fingers, wound infection, known allergy to chitosan and/or polyvinylpyrrolidone, known pregnancy or breast-feeding females, disorders associated with impaired wound healing (e.g., diabetes mellitus), skin diseases in the wound area, and participation in another clinical trial.

#### 2.2.4. Intraoperative Exclusion Criteria

Intraoperatively, patients with avascular fingers or multiple nerve lesions were excluded. Additionally, patients were excluded when tension-free nerve coaptation was possible.

### 2.3. Participants and Randomization

Patients were randomized by the envelope method. The envelopes were numbered based on a randomization list provided by a statistician and assigned to the control group or the treatment group. The randomization was performed with a ratio of 1:1. With a patient defined as eligible for inclusion, the numbered envelope was opened in the operation theatre, and the patient received the nerve repair treatment assigned. Therefore, the patient and the surgeon were blinded to the intended treatment until right before surgical intervention.

### 2.4. Interventions

After preparing the nerve defect on the finger site and ensuring the presence of a nerve gap, patients were allocated to ANG or treated with CNC. ANG was obtained from the medial or lateral cutaneous antebrachial nerve and coaptated end-to-end with 9-0 Ethilon epineural stitches in a retrograde manner. The proximal and distal nerve ends of the intervention group (CNC) were cut back to expose healthy tissue. The CNC was then inserted and fixed to both nerve stumps using 9-0 Ethilon 1–2 epineural sutures, with CNC overlapping the nerve ends by about 2–3 mm (Figure 1). The US Food and Drug Administration has approved CNC as a medical device (K180222). In this study, we used the Reaxon^®^ nerve conduit with a diameter of 2.1 mm (Medovent GmbH, Mainz, Germany).

### 2.5. Follow-Up and Blinding

Follow-up examinations were performed at three, six, and twelve months postoperatively in a blinded manner. Neither the patient nor the examiner were informed about the treatment modality. Two-point discrimination after three, six, and twelve months was used as the primary outcome parameter. A double tip compass evaluated static two-point discrimination at the radial and ulnar fingertip of the injured finger and as the intraindividual control on the uninjured contralateral finger.

For dynamic testing, two points were applied randomly in an axial direction on the injured finger. The two-point discrimination results were categorized using the following grading scale: [0 ≥ 15 mm (poor); 1 = 11–15 mm (fair); 2 = 6–10 mm (good); 3 = 6 mm (excellent). A functional recovery score of 6–10 mm is considered significant.

The assessment with Semmes Weinstein Filaments was administered as a second outcome parameter for the evaluation of the sensory recovery. It was measured with 6 filaments of varying thickness (0 = not testable, 1 = filament 6.65 g (perception of deep pressure), 2 = filament 4.56 g (no protective sensation), 3 = filament 4.31 g (diminished protective sensation), 4 = filament 3.61 g (diminished perception of light touch), 5 = filament 2.83 g (normal perception of light touch)).

Additionally, patients were questioned about their self-experienced pain, which was validated using a visual analogue scale ranging from 0 (no pain) to 10 (heavy pain). Patient satisfaction was determined by interviewing the patient and ranking the results on a scale of 1 (extremely good) to 6 (very bad). The subsequent investigations were conducted by a blinded investigator.

### 2.6. Statistical Analysis

GraphPad Prism V5.0 (GraphPad Software, Inc., La Jolla, CA, USA) software was applied for analysis and drawing. Data of patient age, defect size, regeneration length, and implant length were distributed equally; therefore, an unpaired two-tailed *t*-test was used for analysis. Due to non-normal distribution, a two-tailed Wilcoxon signed-rank test was used to assess two-point discrimination. Values of 3, 6, and 12 months were compared to the baseline results. The two-tailed Wilcoxon signed-rank rest also assessed Semmes Weinstein Filament Testing, patient satisfaction, and pain investigation, comparing three-months postoperative results to results six and 12 months postoperative. The Mann–Whitney U Test was performed for analysis between ANG and CNC at 3, 6, and 12 months postoperatively.

For all analyses, significant findings were defined as those with *p* < 0.05, and a confidence interval of 95% was applied. No post hoc testing was further used. Results were presented with the mean and the standard error of the mean.

## 3. Results

The study enrolled 47 patients. Following randomization, 23 patients were assigned to the ANG group and 24 to the CNC group. Seven patients dropped out of the ANG group and four patients dropped out of the CNC group after three months of follow-up due to non-attendance at follow-up appointments. Four patients dropped out of CNC after six months. Additionally, nine patients in the ANG group and one patient in the CNC group were dropped due to non-attendance at the twelve-month follow-up appointment. The twelve-month follow-up examination was completed by seven patients in the ANG group and fifteen patients in the CNC group (Figure 2).

### 3.1. Qualitative Analysis

The descriptive analysis revealed no significant differences in the patients’ ages (*p* = 0.53). In both groups, males were mostly included in the study. There were no significant differences in the size of the nerve gap between ANG and CNC (*p* = 0.6) nor in the distance between the proximal nerve stump and the finger pulp (regeneration length; *p* = 0.37) (see Table 1).

### 3.2. Primary Outcome Parameter–Tactile Gnosis (Two-Point Discrimination)

Static and dynamic tests for two-point discrimination were conducted immediately after surgery, as well as 3, 6, and 12 months later. For static two-point discrimination, ANG achieved 13.3 ± 1.3 mm after three months, 11.00 ± 2.0 mm after six months, and 7.9 ± 1.1 mm after 12 months. After 12 months, when compared to direct postoperative results (baseline), a significant (*p* < 0.01) improvement in tactile gnosis was observed (Figure 3A).

After three months, the CNC measured 10.73 ± 1.2 mm, 10.3 ± 1.3 mm after six months, and 10.9 ± 1.3 mm after 12 months for static two-point discrimination. After three months (*p* < 0.05), the first significant regeneration occurred, with further significance after 6 (*p* < 0.01) and 12 months (*p* < 0.05) when compared to the direct postoperative results (Figure 3B). No significant differences between ANG and CNC were observed after 3 months (*p* = 0.17), 6 months (*p* = 0.91), or 12 months (*p* = 0.14).

The first significant improvement compared to the baseline with dynamic two-point discrimination was demonstrated after three months for both ANG (*p* < 0.05; 7.4 ± 0.8 mm) and CNC (*p* < 0.01; 7.1 ± 0.8 mm). Improvement was continued in the ANG group after six months with values of 5.0 ± 0.6 mm, while the CNC group maintained its two-point discrimination values (7.8 ± 2.4 mm). After 12 months, ANG demonstrated a steady-state value of 5.3 ± 0.6 mm (*p* < 0.05), whereas CNC showed a slight improvement with 6.6 ± 1.4 mm (*p* < 0.01) (Figure 3C,D). No significant differences were shown in the comparison between the two groups in any of the follow-up examinations (three months: *p* = 0.84, six months: *p* = 0.94, 12 months: *p* = 0.96).

### 3.3. Secondary Outcome Parameter–Functional Analysis by Grading

The results of static and dynamic two-point discrimination were further analyzed by categorizing the functional outcome. The following grading score system was used: 0 ≥ 15 mm (poor); 1 = 11–15 mm (fair); 2 = 6–10 mm (good); 3 = 6 mm (excellent). After three months (*p* = 0.17; 0.6 ± 0.3) and six months (*p* = 0.35; 0.8 ± 0.5) of static grading, ANG demonstrated non-significant results compared to the baseline (*p* = 0.17; 0.6 ± 0.3). After twelve months, the functional outcome for ANG was significantly improved when compared to direct postoperative results, with 2.0 ± 0.2 (*p* ± 0.001). In comparison, CNC demonstrated a significant recovery after three months [1.4 ± 0.3 (*p* < 0.05)] and remained nearly constant after six months at 1.4 ± 0.4 (*p* < 0.05) and twelve months 1.2 ± 0.3 (*p* < 0.05) respectively. After three (*p* = 0.14), six (*p* = 0.37), and twelve (*p* = 0.13) months, there were no significant differences between the CNC and the ANG. For dynamic grading, the first significant improvement was observed after three months in both groups (ANG: 1.9 ± 0.3; CNC: 1.8 ± 0.3) in comparison to baseline (*p* < 0.01). After six months (2.3 ± 0.3) (*p* < 0.01), the ANG continued to improve and then remained close to this level until twelve months postoperatively (2.0 ± 0.4) (*p* < 0.01). CNC also improved significantly after three months when compared to the baseline (*p* < 0.01). After six months, regeneration remained stable (1.5 ± 0.4) (*p* < 0.01) and did not further improve further twelve months after reconstruction (1.8 ± 0.3) (*p* < 0.01). When the ANG and CNC were compared directly, there were no significant differences between the two after three (*p* = 0.95), six (*p* = 0.31), or twelve months (*p* = 0.84). The CNC is close to achieving a good functional outcome with an average tactile gnosis of 6–10 mm twelve months after surgery (Figure 4).

### 3.4. Secondary Outcome Parameter–Semmes Weinstein Monofilament Testing

Semmes Weinstein Monofilament was tested on the injured finger and the uninjured contralateral finger as intraindividual control. Results were categorized into six groups varying from 0 (non-testable) to 5 (normal perception of light touch). ANG presented results of 3.3 ± 0.4 (three months), 4.4 ± 0.2 (six months) and 3.7 ± 0.5 (twelve months). Significant differences were shown compared to intraindividual control were shown for ANG after three months (4.7 ± 0.1; *p* < 0.01). Results after six (4.8 ± 0.1) and twelve months (4.57 ± 0.20) presented no significant differences. For CNC, significant differences compared to intraindividual control were revealed after three (*p* < 0.01; CNC 3.1 ± 0.3 vs. CNC control 4.4 ± 0.2), six (*p* < 0.01; CNC 3.5 ± 0.4 vs. CNC control 4.8 ± 0.1) and twelve months (*p* < 0.05; 3.7 ± 0.4 vs. CNC control 4.8 ± 0.1) (see Figure 5A). No significant differences have been shown. ANG and CNC after three (*p* = 0.73), six (*p* = 0.13) and twelve (*p* = 0.96) months no significant differences have been shown.

### 3.5. Secondary Outcome Parameter–Pain Assessed by the Visual Analogue Scale and Patient Satisfaction

After three months, self-reported pain scores on the visual analogue scale ranged from 2.0 ± 0.9 (ANG) to 1.2 ± 0.5 (CNC). After twelve months ANG showed values of 1.4 ± 1.1 and CNC values of 1.9 ± 0.6 (see Figure 5B). There were no significant differences between ANG and CNC at any of various time points (three months, *p* = 0.66; six months, *p* = 0.52; twelve months, *p* = 0.82).

After three months, patient satisfaction confirmed positive results, with values of 3.1 ± 0.5 (ANG) and 3.8 ± 0.4 (ANG) (CNC). After six months (2.8 ± 0.4 ANG; 2.9 ± 0.3 CNC), patient satisfaction remained high for CNC, but not for ANG (2.3 ± 0.4 ANG; 3.3 ± 0.4 CNC) (Figure 5C). There were no significant differences in patient satisfaction between ANG and CNC during the postoperative evaluation period (three months, *p* = 0.22; six months, *p* = 0.89; twelve months, *p* = 0.09).

### 3.6. Adverse Events

As a direct result of the CNC implementation, one patient developed a wound-healing disorder, which was accompanied by a dislocation and a small fistula of the CNC group. A surgical revision with partial removal of the CNC was required. Following that, the wound healing disorder resolved completely without recurrence. Additionally, wound healing disorders unrelated to the nerve graft or nerve tube were observed in both groups (ANG: 1 vs. CNC: 3) but did not require additional surgical intervention and were conservatively treatable.

## 4. Discussion

Since Millesi et al. published the successful reconstruction of peripheral nerve defects with ANG, no other reconstruction concept was able to show superiority [1,2,14]. At best, artificial nerve conduits or alternative concepts, such as processed allografts of human nerves, led to functional results such as ANG [15]. Due to the reliable, functional outcomes of ANG in peripheral nerve surgery, it is still the gold standard for bridging peripheral nerve defects. However, its disadvantages such as a second operation site with its associated operation risks and the functional loss of the harvested nerve induce a strong need for an alternative in daily clinical practice.

Chitosan with its material properties has the potential to become a serious alternative to well-established ANG. Compatibility to the neural cell lineage [16], the ability for a tailored biodegradation process with non-toxic debris [7], and good mechanical properties [17] have been reported and are the essential requirements for a sufficient bioartificial nerve conduit [8]. First clinical evidence of chitosan’s benefits for PNI showed superior two-point discrimination results to ANG after six months of regeneration by protecting the epineural suture with a CNC [13].

Therefore, we started the first prospective randomized trial to evaluate the potential of this material to bridge peripheral nerve defects. By assessing two-point discrimination as the parameter commonly used for the tactile gnosis [18], we showed an analogous regeneration speed of CNC compared to ANG. Moreover, we first showed significant regeneration compared to baseline within three months (see Figure 3). In line with Neubrech et al. [13], this is the second time chitosan showed beneficial effects on functional recovery, especially in the early phase after PNI. Chitosan’s bioeffective properties shown under in vivo conditions, such as the support of axonal regeneration [10], reduction in scar tissue formation [11,19], or the chitooligosaccharide-induced acceleration of the Schwann Cell cell cycle [20,21], may explain these effects.

Interestingly, CNC showed similar results compared to ANG for two-point discrimination after twelve months (see Figure 3). By focusing on clinically meaningful regeneration, which is achieved with a two-point discrimination of 6–10 mm, CNC also performed comparably to the ANG for dynamic functional regeneration (Figure 4). To our knowledge, this is the first prospective randomized clinical trial that showed the sensory regenerative potential of CNC for small nerve defect sizes in the hand. Even with apparent disadvantages compared to the ANG, such as no cell seeding or an inner architecture for the guidance of the axonal regeneration, CNC can bridge small nerve defects with a proper functional recovery assessed by tactile gnosis. Interestingly, with the convincing regeneration of CNC within the first three months, especially for static two-point discrimination, further improvement seems to be limited for the rest of the regeneration period. ANG, in contrast, showed constant improvements during the regeneration period and showed slightly better results than CNC after 12 months. This could be due to the influences of chitosan biodegradation or the natural inner architecture of ANG compared to the hollow design of CNC. However, this requires further investigation seeing as there can be various explanations.

Furthermore, the Semmes Weinstein Filament Test improved for CNC and ANG during regeneration. However, after six months, ANG showed better functional results than 12 months postoperatively (see Figure 5A). This may be reasoned by postoperative nerve compression due scar tissue developments for ANG. In contrast, chitosan’s scar-preventing abilities may hide this effect for CNC. However, these interpretations are limited due to the small population size, particularly for ANG, and this result may also be based on a small population bias.

The prospective and randomized study design allows us to gain reliable data on potential chitosan nerve conduits in a future clinical setting.

Nonetheless, nerve conduit studies, in general, suffer from a lack of unity considering clinical models, investigated nerves, and assessment methods of functional recovery. The clinical model used in this study focuses on the hand’s sensory nerves and excludes motoric and mixed nerves. This restricts the evaluation to merely sensory recovery, yet this model is comparable to future research. As mentioned by others, the main assessment for functional analysis should be usingtwo-point discrimination as the benchmark for tactile gnosis and for evaluating performances in the context of others [22].

There is a huge variety of materials and concepts described in the literature, which also present the possibility to bridge peripheral nerve defects in the hand [23]. FDA approved materials based on collagen (Neuragen^®^) [24], co-polymer of lactide and caprolactone (Neurolac^®^) [25], polyglycolic acid (Neurotube^®^) [26], or processed nerve allografts [27] have been described in the literature. Due to the study’s heterogeneity of gap lengths, regeneration period, patient collective, and assessment methods, a direct comparison is hard to conduct. Despite the variation in the performance of the static two-point discrimination [28], it appears to be one of the most reliable factors for comparing the functional capacity of nerve conduits. For lactide and caprolactone-based nerve conduits (Neurolac^®^), Chiriac et al. showed static two-point discrimination of 24.9 mm (range from 6 to 30 mm) for an average nerve defect in the upper extremity of 11.9 mm [25]. These results can only barely be compared because of the considerable heterogeneity of the different lesion sides (arm, elbow, forearm, wrist, palm, and fingers included), nerve types, and an extended regeneration period with an average of 21.9 months. However, in the context of eight complications in a case series of 28 nerve lesions treated with a polyDL-lactide-εcaprolactone) nerve conduit, CNC seems to be a proper alternative considering safety and performance. In the prospective randomized clinical trial by Rinker et al., a polyglycolic acid conduit and a vein conduit were used to bridge extended digital nerve lesions (≥10 mm). The main assessment parameter was two-point discrimination, and due to the same clinical model and primary outcome parameter, this study showed high comparability to our research. For a polyglycolic acid nerve conduit (Neurotube^®^), static two-point discriminations of 9.6 ± 1.9 mm and 9.3 ± 1.9 mm for vein conduits after a 12-month regeneration period were revealed [26]. Compared to our study, with results of 10.9 ± 1.3 for an average gap length of 13.3 ± 1.9 mm, CNC seemed to have comparable outcomes relative to this FDA-approved nerve conduit. Moreover, in contrast to acid-based materials, such as polyglycolic acid or polylactide derivates [23], chitosan seems not to impair peripheral nerve regeneration by a pH decrease or show signs of inflammatory foreign body reaction but rather to support it by its neuroprotective metabolites during biodegradation [29]. Recently, processed nerve allografts have shown equivalent results to ANG [15]. Due to its inner architecture and minimal immunogenicity, processed nerve allografts come close to the primary objective of nerve conduits research to mimic the physiological nerve. Safa et al. demonstrated a significant motor and sensory recovery for regeneration up to 70 mm nerve defect for processed nerve allografts. Nonetheless, the study design includes sensory, motoric, and mixed nerves on all levels of the upper extremity [27], and the follow-up was conducted 779 days on average postoperatively. In the subgroup analysis for digital nerve repair, 84 % of the patient has achieved a meaningful recovery (S3) defined as a return of pain and tactile sensibility with a static two-point discrimination > 15 mm [18]. Despite the more extended average gap size of 21 ± 12 mm for sensory nerves, processed nerve allografts do not appear to provide superior functional recovery results compared to CNC. CNC’s disadvantages, such as a missing inner layer, may be compensated by beneficial material properties such as preventing scar tissue [11] or neuroma formation [12], which have not been described for processed nerve allografts.

In line with this, we have not observed signs of a painful neuroma development for twelve months after bridging PNI with CNC similarly to others using different materials [30,31,32]. Usually, the development of a painful neuroma after PNI in the hand is a common problem and appears within six to twelve months after injury [33]. With values of 1.9 ± 0.6 for CNC after twelve months in the visual analog scale (Figure 5B), reliable patient satisfaction (Figure 5C), and no surgical revision due to pain, the development of a painful neuroma formation seems to be no issue in the application of CNC for bridging peripheral nerve defects. Supported by Neubrech et al., who also showed no painful neuroma formation 13, chitosan’s neuroma preventing effects under in vivo condition 12 also seem to be assignable in a clinical setting.

This is the first clinical study to evaluate chitosan’s ability to bridge peripheral nerve defects. However, it is limited by the study’s small sample size and high drop-out rate during the twelve-month follow-up. Consequently, the current study suffers from a selection bias by focusing on patients with postoperative issues and a lack of information on adverse events with a low incidence. Furthermore, due to the small population size for ANG and CNC, the results provide only a hint for the future applications of chitosan nerve conduits. ANG had a drop-out rate of 16 patients, whereas CNC had a drop-out rate of nine patients. ANG’s results may be underpowered by the small population size and the assumption that a patient with no complaints is not encouraged to attend follow-up sessions. However, postoperative pain measured by VAS does not support this assumption by showing nearly the same VAS score for both groups.

Future studies should focus on the non-inferiority testing of CNC against the ANG based on a higher population size.

We limited our assessment to sensory nerve recovery to obtain reliable data comparing the control (ANG) and intervention groups (CNC). The study does not address the potential of CNC for motoric or mixed sensory-motoric nerves or for more extended lesion sites, which also should be addressed in the future. The supplementary seeding of CNC with cells and growth factors may be another approach to improve peripheral nerve regeneration. Exemplary mesenchymal stromal cells [34,35], adipose-derived stem cells [36], or plated rich plasma [37,38] may support nerve regeneration in combination with a CNC without having the disadvantage of functional loss at the donor site.

In conclusion, CNC supported peripheral nerve regeneration in a clinical setting. Patients treated with CNC presented appropriate tactile gnosis within the first six months and similar results to the ANG after twelve months without having signs of any painful neuroma development. Axonal regeneration facilitated by the hollow architecture and the material itself may represent a significant step forward in the future treatment of PNI. However, the non-inferiority testing of CNC compared to ANG is yet to come in the future. We recommend additional clinical trials to evaluate the potential of CNC in peripheral nerve surgery, with a particular emphasis on long-term results, biodegradation, and application relative to motoric or mixed motoric nerves. In the future, CNC may be a viable option for bridging small sensory nerve defects in the hand.

## Figures and Tables

**Figure 1 jpm-12-00900-f001:**
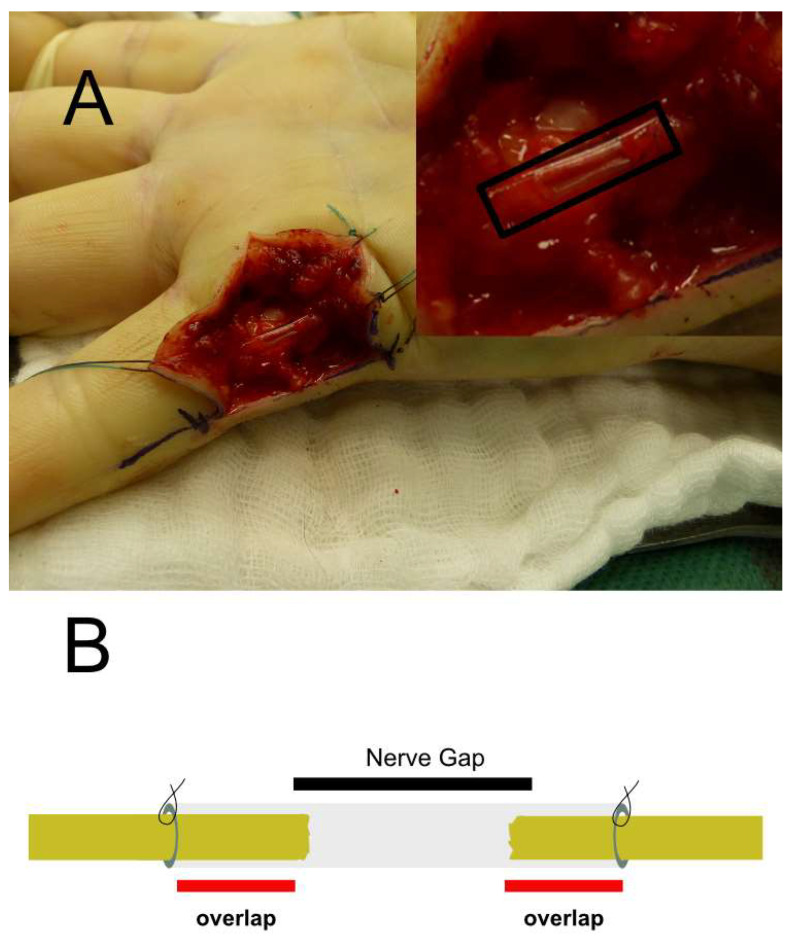
**Analysis of the lesion side after finger trauma.** Patient with a peripheral nerve defect of the ulnar nerve of the little finger (**A**) and bridging by a CNC. In this case, a peripheral nerve defect of approximately 9 mm is bridged by the CNC (Magnification A 2×). Schematic illustration of the tubulization technique for bridging the peripheral nerve defect with the CNC. Note the 2–3 mm overlap of CNC proximal and distal to allow guided sprouting of the axons via the conduit (**B**).

**Figure 2 jpm-12-00900-f002:**
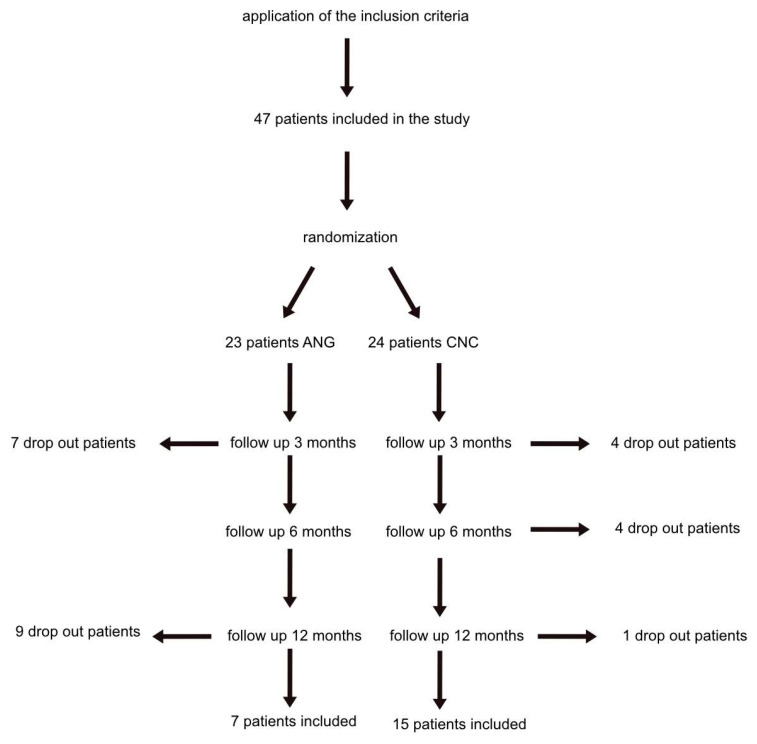
Overview of the patient inclusion during the study and a regeneration period of twelve months.

**Figure 3 jpm-12-00900-f003:**
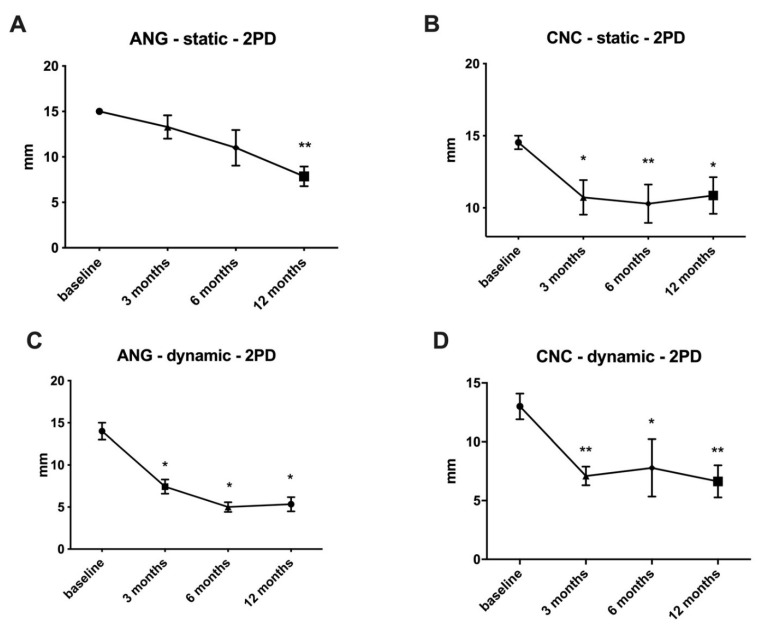
**Two-point discrimination for ANG and CNC after three, six, and twelve months.** Static two-point discrimination showed the first significant recovery compared to baseline after twelve months for the ANG group (**A**; *p* < 0.01). In contrast, CNC presented the first significant recovery compared to the baseline after three months (**B**; *p* < 0.05). Dynamic two-point discrimination revealed slightly superior results compared to static two-point discrimination, with the first significant results for ANG (**C**; *p* < 0.05) and CNC (**D**; *p* < 0.01) after three months. ANG: autologous nerve graft; CNC: chitosan nerve conduit; * *p* < 0.05, ** *p* < 0.01.

**Figure 4 jpm-12-00900-f004:**
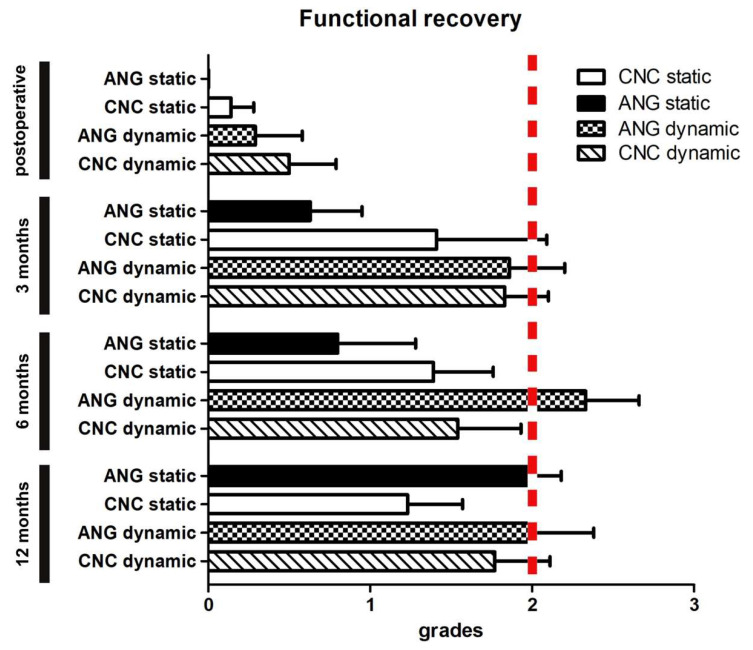
**Direct comparison of ANG and CNC focusing on functional recovery.** Categorizing the two-point discrimination showed that meaningful regeneration is shown after twelve months by the ANG and nearly similar for CNC. Interestingly, dynamic regeneration showed superior results to static analysis focusing on functional recovery.

**Figure 5 jpm-12-00900-f005:**
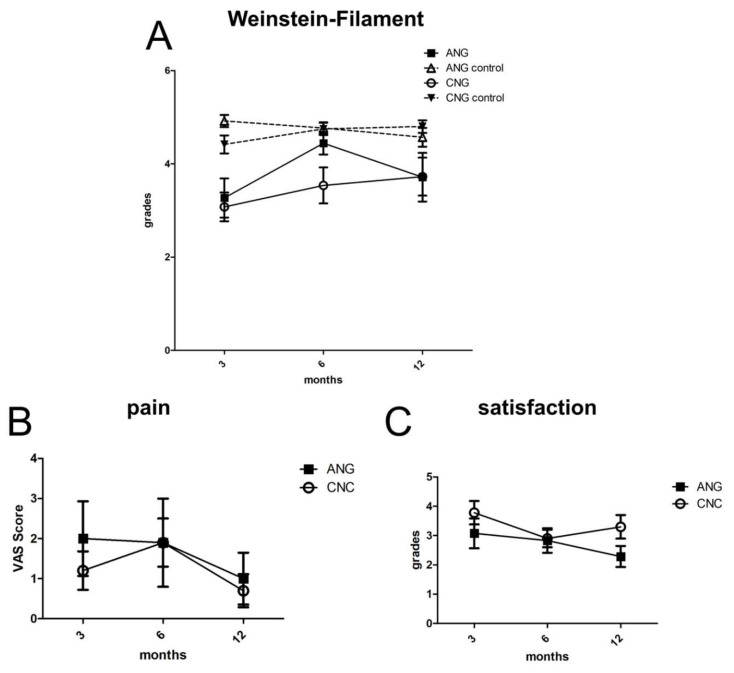
**Semmes Weinstein Monofilament Test, pain assessed by the VAS score, and patient satisfaction.** Semmes Weinstein Monofilament Test exposed no significant differences between ANG and CNC over the regeneration period. After twelve months of regeneration, CNC showed similar clinical results compared to ANG (**A**). Neither ANG nor CNC severe pain was assessed (**B**). Furthermore, reasonable patient satisfaction was seen after six and twelve months without presenting significant differences between ANG and CNC (**C**).

**Table 1 jpm-12-00900-t001:** **Patient collective and qualitative analysis.** Both groups showed no statistically significant differences in patient age, peripheral nerve defect distance, regeneration length (the distance between the proximal nerve lesion and the finger pulp), and implant length. CNC: chitosan nerve conduit; ANG: autologous nerve graft.

		ANG	CNC	*p*-Value
Age		45.43 ± 5.85	35.93 ± 3.33	0.5331
Sex				
	male	86% (6)	67% (10)	
	female	14% (1)	33% (5)	
Injury				
	crush	29% (2)	20% (3)	
	cut	71% (5)	73% (11)	
	both	0% (0)	7% (1)	
Dominant hand				
	right	100% (7)	80% (12)	
	left	0% (0)	13% (2)	
Injury side				
	right	43% (3)	60% (9)	
	left	57% (4)	40% (6)	
	dominant	43% (3)	47% (7)	
Defect size (mm)		12.80 ± 2.01	13.25 ± 1.89	0.6218
Regeneration length (mm)		56.10 ± 0.76	66.88 ± 0.81	0.3653
Implantat length (mm)		15.00 ± 2.76	17.38 ± 2.05	0.6746

## Data Availability

The data presented in this study are available on request from the corresponding author. The data are not publicly available due to privacy reasons.

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
