# Peer review of "Closing the Gap: Bridging Peripheral Sensory Nerve Defects with a Chitosan-Based Conduit a Randomized Prospective Clinical Trial"

_jpm, 2022, doi:10.3390/jpm12060900_

Round 1

Reviewer 1 Report

Dear authors,
I’ve reviewed the manuscript "Closing the gap: bridging peripheral sensory nerve defects with a chitosan-based conduit. A randomized prospective clinical trial." I would like to congratulate the authors for providing an interesting and well planed study.

However, I would like to share my points of critique with you:

Major concerns:
For the evaluation of the two-point discrimination (compared with baseline) Kruskal Wallis was used. In this case, we have a dependent sample (dependent factor "time") and one must therefore take a non-parametric test for dependent measurements (e.g. Friedman Test). For intraindividual comparisons tests with repeated measures must be used (Friedman test, Wilcoxon test), for interindividual comparisons tests without repeated measures (Mann-Whitney-U test, Kruskal-Wallis test).  Furthermore, the statistics part does not describe a post-hoc test and does not show which values are used for the descriptive representation (e.g. standard deviation, mean, interquartile range, median; e.g. line 239, 264). 

In the discussion, the results between CNC and ANG are described as similar, although comparative tests were used. To examine "equivalent" results, other methods are used (for example, non-inferiority trial). More precise wording should be used and e.g. non-inferiority studies can be suggested as an outlook for further studies. Depending on what the goal of the study is, a different method is used: is the new conduit as good as ANG (non-inferiority study) or better (comparative test as in this study). I recommend that the statistical analyses be reviewed and corrected by a statistician.

The discussion is in its current form too superficial. I believe there are many aspects that can be discussed (e.g. figure 6 Weinstein-Filament and pain: discrepancy at 6 months compared to 3 and 12 months; higher drop out / less patients after 12 months in the ANG group and consequences for the study). More comprehensive integration of the own results and presentation of advantages and disadvantages of the study design compared to the other cited studies would improve the discussion section. 

Minor concerns:

Line 99: list of participating centers in this multi-center study

Line 113: add time span for recent trauma (definition for recent)

Line 132: Patients sections in methods and results are described very superficially. Parameters such as etiology, time of trauma etc. are missing. If possible add these parameters (incl. age, sex, regeneration length, defect sice) in a table with statistical comparisons between the two groups (ANG vs. CNC). For a better overview, i recommend to present the information of figure 3 in a table.

Line 133: please specify more clearly the randomization process (randomization with a software, website?)

Line 202: a few p-values are not presented correctly.

Line 273: In this chapter the results are not presented correctly (e.g. missing +/-)

Line 242: PREoperative results? 

Line 322: please specify "SC".

I would recommend a review by a native English speaker.

Author Response

Closing the gap: bridging peripheral sensory nerve defects with a chitosan-based conduit. A randomized prospective clinical trial.

Dear Reviewer,

Thank you very much for your substantial revision considering our paper. We really appreciate the efforts you have put into to improve this manuscript. In the following, I will answer your questions in detail. 

For the evaluation of the two-point discrimination (compared with baseline) Kruskal Wallis was used. In this case, we have a dependent sample (dependent factor "time") and one must therefore take a non-parametric test for dependent measurements (e.g. Friedman Test). For intraindividual comparisons tests with repeated measures must be used (Friedman test, Wilcoxon test), for interindividual comparisons tests without repeated measures (Mann-Whitney-U test, Kruskal-Wallis test).  Furthermore, the statistics part does not describe a post-hoc test and does not show which values are used for the descriptive representation (e.g. standard deviation, mean, interquartile range, median; e.g. line 239, 264). 

Thank you very much for your comment.  

We agree that Kruskal Wallis is not a suitable statistic test for answering the significance of the two-point discrimination between the baseline and the different periods. Indeed, time is the dependent factor, so we changed the statistical testing. Since the data is not distributed normally, we used a two-tailed Wilcoxon matched-pair signed-rank test for statistical analysis of the two-point discrimination. We tested the two-point discrimination with the baseline results and the results 3, 6 and 12 months postoperatively.    

The statistical analysis for the two-point discrimination is clarified under materials and methods by adding the following sentences:

Graphics and results were adjusted in the results and discussion section.

We also modified the part "statistical analysis" in the material and methods section in the following way: 

" GraphPad Prism V5.0 (GraphPad Software, Inc., La Jolla, Calif.) software was applied for analysis and drawing. Data of patient age, defect size, regeneration length and implantat length were distributed equally, and therefore an unpaired two-tailed t-test was used for analysis. Due to non-normal distribution, a two-tailed Wilcoxon signed-rank test was used. Values of 3, 6 and 12 months were compared to the baseline results. The two-tailed Wilcoxon signed-rank rest also assessed Semmes Weinstein Filament Testing, patient satisfaction, and pain investigation, comparing results three months postoperative to results six and 12 months postoperative. The Mann-Whitney U-Test was performed for analysis between ANG and CNC at the three, six, and twelve months postoperatively.

For all analyses, significant findings were defined as those with p < 0.05 and a confidence interval of 95% was applied. No post-hoc testing was applied. Results were presented with the mean and the standard error of the mean."

In the discussion, the results between CNC and ANG are described as similar, although comparative tests were used. To examine "equivalent" results, other methods are used (for example, noninferiority trial). More precise wording should be used and e.g. noninferiority studies can be suggested as an outlook for further studies. Depending on what the goal of the study is, a different method is used: is the new conduit as good as ANG (noninferiority study) or better (comparative test as in this study). I recommend that the statistical analyses be reviewed and corrected by a statistician.

Thank you very much for this comment. 

The Reviewer is correct. We should clarify that non-noninferiority testing was demonstrated. However, this study is the first human study showing the potential of a chitosan nerve conduit in a prospective randomized clinical trial.

This study aimed to show that the chitosan nerve conduit may be a feasible alternative to the ANG in the future. Unfortunately, due to the small population size and the minor differences between the CNC in the ANG, proper noninferiority testing is not possible, as mentioned by the statistician. Therefore, we only can talk about similar results between ANG and CNC and not about equal results in the discussion section. We adjusted the following sentences in the discussion part to take your advice into account.

For further clarification, we added a sentence considering the noninferiority testing in the limitations and clarified that the results only give the hint for a possible alternative in the future. Furthermore, we adjusted in the discussion section focusing on the wording between the ANG and CNC for further clarification. 

"Due to the small population size for ANG and CNC the results give a hint for the future application of chitosan nerve conduits. Future studies should focus on the noninferiority testing of the CNC against the ANG based on a higher population size. ANG had a drop-out rate of 16 patients, whereas CNC had a drop-out rate of seven patients. ANG's results may be underpowered by the small population size and the assumption that a patient with no complaints is not encouraged to attend follow-up sessions. However, postoperative pain measured by the VAS does not support this assumption by showing nearly the same VAS score."

The discussion is in its current form too superficial. I believe there are many aspects that can be discussed (e.g. figure 6 Weinstein-Filament and pain: discrepancy at 6 months compared to 3 and 12 months; higher drop out / less patients after 12 months in the ANG group and consequences for the study). More comprehensive integration of the own results and presentation of advantages and disadvantages of the study design compared to the other cited studies would improve the discussion section. 

Thank you for your comment. 

 We extended the discussion with an additional focus on the results of the Weinstein-Filament Testing by adding the following passage:

" Furthermore, the Semmes Weinstein Filament Testing improved for CNC and ANG during regeneration. However, after six months, ANG showed better functional results than 12 months postoperatively [see figure 5A]. This may be reasoned by postoperative nerve compression due scar tissue development for ANG. In contrast, Chitosan’s scar preventing abilities may hide this effect for CNC. However, these interpretations are limited due to the small population size, in particular for ANG, and this result may also base on a small population bias.”

We further go more into detail focusing on our study compared to studies with a similar design. 

The prospective and randomized study design allows us to gain reliable data on potential chitosan nerve conduits in a future clinical setting.

Nonetheless, nerve conduit studies, in general, suffer from the lack of unity considering clinical models, investigated nerves, and assessment methods of functional recovery. The clinical model used in this study focuses on the hand's sensory nerves and excludes motoric and mixed nerves. This restricts the evaluation to merely sensory recovery, yet this model is comparable to future research. As mentioned by others, the main assessment for functional analysis should be the two-point discrimination as the benchmark for the tactile gnosis and to evaluate performance in the context of others 23.

 There is a huge variety of materials and concepts described in the literature, which also present the possibility to bridge peripheral nerve defects in the hand 24. FDA approved materials based on collagen (Neuragen) 25, co-polymer of lactide and caprolactone (Neurolac) 26, polyglycolic acid (Neurotube) 27 or processed nerve allografts 28 have been described in the literature. Due to the study's heterogeneity of gap lengths, regeneration period, patient collective, and assessment methods, a direct comparison is hard to make. Despite the variation in the performance of the static two-point discrimination 29, it seems one of the most reliable factors for comparing the functional capacity of nerve conduits. For lactide and caprolactone-based nerve conduits (Neurolac), Chiriac et al. showed static two-point discrimination of 24.9 mm (range from 6 to 30 mm) for an average nerve defect in the upper extremity of 11.9 mm 26. These results can only barely be compared because of the considerable heterogeneity of the different lesion sides (arm, elbow, forearm, wrist, palm and fingers included), nerve types and an extended regeneration period with an average of 21.9 months. However, in the context of eight complications in a case series of 28 nerve lesions treated with a polyDL-lactide-εcaprolactone) nerve conduit, CNC seems to be a proper alternative considering safety and performance. In the prospective randomized clinical trial by Rinker et al., a polyglycolic acid conduit and a vein conduit were used to bridge extended digital nerve lesions (> 10mm). Main assessment parameter was the two-point-discrimination, and due to the same clinical model and primary outcome parameter, this study showed high comparability to our research. For a polyglycolic acid nerve conduit (Neurotube) static two-point discrimination of 9.6 ± 1.9 mm and 9.3  ± 1.9 mm for vein conduits after 12 months regeneration period were revealed 27. Compared to our study, with results of 10.9 ± 1.3 for an average gap length of 13.3 ± 1.9 mm, CNC seemed to have comparable outcomes to this FDA-approved nerve conduit. Moreover, in contrast to acid-based materials, like polyglycolic acid or polylactide derivates 24, Chitosan seems not to impair peripheral nerve regeneration by a pH decrease or signs of inflammatory foreign body reaction but rather to support it by its neuroprotective metabolites during biodegradation 30. Recently, processed nerve allografts have shown equivalent results to the ANG15. Due to its inner architecture and minimal immunogenicity, processed nerve allografts come close to the primary objective of nerve conduits research to mimic the physiological nerve. Safa et al. demonstrated a significant motor and sensory recovery for regeneration up to 70 mm nerve defect for processed nerve allografts. Nonetheless, the study design includes sensory, motoric, and mixed nerves on all levels of the upper extremity28, and the follow-up was conducted 779 days on average postoperatively. In the subgroup analysis for digital nerve repair,  84 % of the patient has achieved a meaningful recovery (S3) defined as a return of pain and tactile sensibility with astatic two-point discrimination > 15mm 18. Despite the more extended average gap size of 21 ± 12 mm for sensory nerves, processed nerve allografts do not appear to provide superior functional recovery results compared to CNC. CNC’s disadvantages, like the missing inner layer, may be compensated by the beneficial material properties like preventing scar tissue 11 or neuroma formation 12, which not have been described for processed nerve allografts.”

Minor concerns:

Line 99: list of participating centers in this multi-center study

The centers were added to the Materials and Methods Section

"The study was conducted in the trauma centers of the BG Trauma Center Bermannstrost in Halle (Germnay) and the University Hospital Schleswig-Holstein in Lübeck (Germany). "

Line 113: add time span for recent trauma (definition for recent)

The sentence was adjusted by the recommendation of the Reviewer. 

"Preoperative inclusion criteria for patients were as follows: lesion between the distal border of the carpal tunnel and the distal finger joint, age between 18 and 65 years, trauma within a time period of six months (including acute trauma), signed informed consent, and clinically complete loss of nerve-related sensitivity."

Line 132: Patients sections in methods and results are described very superficially. Parameters such as etiology, time of trauma etc. are missing. If possible add these parameters (incl. age, sex, regeneration length, defect sice) in a table with statistical comparisons between the two groups (ANG vs. CNC). For a better overview, i recommend to present the information of figure 3 in a table.

Thanks for the comment, the removed the figure and add the following table for more clarity.

Line 133: please specify more clearly the randomization process (randomization with a software, website?)

Thanks for the advice. The Reviewer is correct that the description of the randomization has to be described more appropriately. 

Therefore, we changed the passage of the manuscript in the following way 

"Patients were randomized by the envelope method. The envelopes were numbered based on a randomization list provided by a statistician and assigned to the control group or the treatment group based on a randomization list provided by a statistician. The randomization was performed with a ratio of 1:1. With a patient defined as eligible for inclusion, the numbered envelope was opened in the operation theatre, and the patient received the nerve repair treatment assigned. Therefore, the patient and the surgeon were blinded to the intended treatment until right before the surgical intervention." 

Line 202: a few p-values are not presented correctly.

p- values were adjusted in the manuscript. 

Line 273: In this chapter the results are not presented correctly (e.g. missing +/-)

Thanks for the advice, results have been adjusted. 

Line 242: PREoperative results? 

Term was changed to "direct postoperative results". Thanks for the advice.

Line 322: please specify "SC".

Thanks for the advice. SC was corrected into Schwann cell 

I would recommend a review by a native English speaker.

The manuscript was reviewed by a native English speaker and adjusted in all settings due to her recommendations. 

Reviewer 2 Report

This study aimed to demonstrate the regeneration potential of chitosan-based nerve conduits in digital nerve defects compared to the current gold standard, the ANG.

Authors conclude that Chitosan-based nerve conduits are safe and suitable to bridge nerve lesions up to 26 mm in the hand. Tactile gnosis improved significantly during the early regeneration period, and functional outcomes were similar to those obtained with an autologous nerve graft. Thus, Chitosan appears to be a sufficient substitute for autologous nerve grafts in the treatment of small nerve defects in the hand.

The paper is well written, and the methodology is good. However, I have some doubt that have to be clarified:

1)How the authors decides that intraoperative gap must be less than 30 mm in diameter? You have to specify this assumption

2)Patients age between 18 and 65 years: Please justify this inclusion criterion

3)Randomization:

With a patient defined as eligible for inclusion, the randomization envelope was opened, and the patient received the nerve repair treatment assigned. Randomization was performed with a ratio of 1:1.: Please specify who gave the envelope, a blind physician? Blind surgeon?

4) In the discussion you can add that in addition PrP could play a role in the nerve regenerative process, PrP could be tested and could be added in the biocompatible tubes 

citing these articles:

Hersant  B, SidAhmed-Mezi M, Aboud  C,  Niddam J, Levy S, Mernier T, La Padula S, Meningaud JP.

Synergistic Effects of Autologous Platelet-Rich Plasma and Hyaluronic Acid Injections on Facial Skin Rejuvenation. Aesthet Surg J. 2021 Jun 14;41(7):NP854-NP865.

La Padula S, Hersant B, Pizza C, Chesné C, Jamin A, Ben Mosbah I, Errico C, D'Andrea

F, Rega U, Persichetti P, Meningaud JP. Striae Distensae: In Vitro Study and Assessment

of Combined Treatment With Sodium Ascorbate and Platelet-Rich Plasma on

Fibroblasts. Aesthetic Plast Surg. 2021 Jun;45(3):1282-1293.

La Padula S, Hersant B, Meningaud JP. Intraoperative use of indocyanine green

angiography for selecting the more reliable perforator of the anterolateral thigh flap: A

comparison study. Microsurgery. 2018 Oct;38(7):738-744.

Author Response

Revision - Closing the gap: bridging peripheral sensory nerve defects with a chitosan-based conduit. A randomized prospective clinical trial.

Dear Reviewer, 

Thank you very much for your advices and your time to improve our work. 

In the following, we adjusted the manuscript based on your recommendations. 

1)How the authors decides that intraoperative gap must be less than 30 mm in diameter? You have to specify this assumption

Thanks for the advice. The diameter is not the correct term, we meant that the nerve size length must be less than 30 mm 

2)Patients age between 18 and 65 years: Please justify this inclusion criterion

Thanks for the advice.

To gain the first data in humans, we decided to exclude children or teenagers from our study. We want not to be compromised by the speciality in physiology or better regeneration potential. CNC has to work as a conduit for an average aged population. Patients over 65 may have a higher incidence of preconditions, which may also influence the CNC results. 

However, the Reviewer is correct; future studies should also include patients under 18 and over 65 years of age.  

3)Randomization:

Thanks for the advice. The Reviewer is correct, that the description of the randomization has to be described more appropriately. 

Therefore, we changed the passage of the manuscript in the following way 

"Patients were randomized by the envelope method. The envelopes were numbered based on a randomization list provided by a statistician and assigned to the control group or the treatment group based on a randomization list provided by a statistician. The randomization was performed with a ratio of 1:1. With a patient defined as eligible for inclusion, the numbered envelope was opened in the operation theatre, and the patient received the nerve repair treatment assigned. Therefore, the patient and the surgeon were blinded to the intended treatment until right before the surgical intervention." 

4) In the discussion you can add that in addition PrP could play a role in the nerve regenerative process, PrP could be tested and could be added in the biocompatible tubes 

citing these articles:

Hersant  B, SidAhmed-Mezi M, Aboud  C,  Niddam J, Levy S, Mernier T, La Padula S, Meningaud JP.

Synergistic Effects of Autologous Platelet-Rich Plasma and Hyaluronic Acid Injections on Facial Skin Rejuvenation. Aesthet Surg J. 2021 Jun 14;41(7):NP854-NP865.

La Padula S, Hersant B, Pizza C, Chesné C, Jamin A, Ben Mosbah I, Errico C, D'Andrea

F, Rega U, Persichetti P, Meningaud JP. Striae Distensae: In Vitro Study and Assessment

of Combined Treatment With Sodium Ascorbate and Platelet-Rich Plasma on

Fibroblasts. Aesthetic Plast Surg. 2021 Jun;45(3):1282-1293.

We added both references in the manuscript by adding the following passage.

" The supplementary seeding of CNC with cells and growth factors may be another approach in the future to improve peripheral nerve regeneration. Exemplary mesenchymal stromal cells34,35, differentiated adipocytes36, or plated rich plasma37,38 may support nerve regeneration in combination with a CNC without having the disadvantage of functional loss at the donor site". 

Round 2

Reviewer 1 Report

The paper has been significantly improved due to the revision.